# Phytochemical Characterization, Antioxidant Activity, and Cytotoxicity of Methanolic Leaf Extract of *Chlorophytum Comosum* (Green Type) (Thunb.) Jacq

**DOI:** 10.3390/molecules27030762

**Published:** 2022-01-24

**Authors:** Igor V. Rzhepakovsky, David A. Areshidze, Svetlana S. Avanesyan, Wolf D. Grimm, Natalya V. Filatova, Aleksander V. Kalinin, Stanislav G. Kochergin, Maria A. Kozlova, Vladimir P. Kurchenko, Marina N. Sizonenko, Alexei A. Terentiev, Lyudmila D. Timchenko, Maria M. Trigub, Andrey A. Nagdalian, Sergei I. Piskov

**Affiliations:** 1Faculty of Medicine and Biology, North-Caucasus Federal University, Pushkina Street 1, 355000 Stavropol, Russia; 78igorr@mail.ru (I.V.R.); s.avanesan@yandex.ru (S.S.A.); kurchenko@tut.by (V.P.K.); risha_veresk@mail.ru (M.N.S.); l_timchenko@mail.ru (L.D.T.); piskovsi77@mail.ru (S.I.P.); 2Laboratory of Cell Pathology, Research Institute of Human Morphology, Russian Academy of Medical Science, Tsyurupa Street 3, 117418 Moscow, Russia; labcelpat@mail.ru (D.A.A.); ma.kozlova2021@outlook.com (M.A.K.); 3Institute of Problems of Chemical Physics, Russian Academy of Sciences (IPCP RAS), Academician Semenov Avenue 1, 142432 Chernogolovka, Russia; natasha55555@yandex.ru (N.V.F.); alexei@icp.ac.ru (A.A.T.); trigub.m@gmail.com (M.M.T.); 4Faculty of Health, Witten/Herdecke University, Alfred-Herrhausen-Straße 50, 58448 Witten, Germany; prof-wolf.grimm@yahoo.de; 5Stavropolsky Antiplague Scientific Research Institute (SAPRI), Sovetskaya Street 13/15, 355106 Stavropol, Russia; ruslankalmykov777@yandex.ru; 6InCome Ltd., Tukhachevskogo Street 21, 355040 Stavropol, Russia; wwwstas@yandex.ru; 7Belarusian State University, Nezavisimosti Avenue 4, 220030 Minsk, Belarus; 8Medicinal Chemistry Research and Education Center, Moscow Region State University, Very Voloshinoi Street, 141014 Mytishchi, Russia; 9Saint Petersburg State Agrarian University, Peterburgskoye Shosse 2, 196605 Saint Petersburg, Russia

**Keywords:** *Chlorophytum comosum*, antioxidant activity, cytotoxicity, herbal medicine, biological active substances

## Abstract

*Chlorophytum* genus has been extensively studied due to its diverse biological activities. We evaluated the methanolic extract of leaves of *Chlorophytum comosum* (Green type) (Thunb.) Jacques, the species that is less studied compared to *C. borivilianum*. The aim was to identify phytoconstituents of the methanolic extract of leaves of *C. comosum* and biological properties of its different fractions. Water fraction was analyzed with matrix-assisted laser desorption/ionization time-of-flight (MALDI-TOF) mass spectrometry. Nineteen compounds belonging to different chemical classes were identified in the methanolic extract of leaves of *C. comosum* (Green type) (Thunb.) Jacques. In addition to several fatty acids, isoprenoid and steroid compounds were found among the most abundant constituents. One of the identified compounds, 4′-methylphenyl-1C-sulfonyl-β-d-galactoside, was not detected earlier in *Chlorophytum* extracts. The water fraction was toxic to HeLa cells but not to Vero cells. Our data demonstrate that methanolic extract of leaves of *C. comosum* can be a valuable source of bioactive constituents. The water fraction of the extract exhibited promising antitumor potential based on a high ratio of HeLa vs. Vero cytotoxicity.

## 1. Introduction

*Chlorophytum comosum* belongs to the genus *Chlorophytum* that covers more than 200 species [1,2,3]. A number of *Chlorophytum* species are referred to as medicinal herbs in the traditional medicine of India [4,5], China [6,7] and Africa [8]. Several species are known as a source of ‘Safed musli’, a major ingredient of herbal preparations with immunomodulatory, adaptogenic, aphrodisiac and other activities [2,3,4,9,10].

Due to its long use in history in traditional medicine, the *Chlorophytum* genus has drawn the attention of researchers in evidence-based medicine. Preparations obtained from different *Chlorophytum* species have been studied for their biological activity and were found to possess immunomodulatory and anti-infectious [11,12,13,14,15,16], antibacterial [17], antinociceptive [18] and antioxidant activities [19,20,21,22,23,24,25,26] to improve male sexual health [27,28,29,30,31,32,33,34], to ameliorate manifestations of diabetes, hyperglycemia and hyperlipidemia [35,36,37,38] or toxic hepatic and testicular impairments [20,39,40].

The antitumor potential of the *Chlorophytum* genus has also been studied. Most studies have been performed in vitro with the use of herbal extracts, their fractions or purified constituents. Butanol extract of *C. comosum* roots was shown to inhibit proliferation and induce apoptosis in four cell lines, mostly of hematological origin. Corresponding to the provided data on the concentrations used in the study, the antiproliferative effect of the extract for T-cell leukemia CCRF-HSB-2 cells exceeded that of Actinomycin D 7- to 8-fold [41]. Crude methanol/dichloromethane extract and butanol (saponin) fraction of methanolic extract of *C. borivilianum* roots have been demonstrated to have relatively low toxicity with IC50 > 100 mkg/mL to three solid tumor cell lines [22]. Similar data were obtained for promyelocytic leukemia HL-60 cells treated with methanolic extracts or saponin fractions of roots of five *Chlorophytum* species [4].

Water extract of *C. borivilianum* roots was examined for antitumor activity in vivo. It has been shown that the extract suppressed the onset and progression of skin tumors induced by carcinogen/promoter application [25].

Phytochemistry studies have revealed compounds of various chemical classes to present in *Chlorophytum* plants. Both common and specific constituents have been isolated from underground parts of *Chlorophytum* species with the use of different extraction and fractionation methods [17,42,43,44,45]. The major class of phytochemicals of the *Chlorophytum* genus is represented by a versatile group of saponins that exert different biological activities [1,37,46,47,48,49,50,51,52].

In vitro cytotoxicity, studies were carried out with saponins purified from butanol fractions obtained from methanolic extracts of underground parts of *C. comosum* [46,47] and rhizomes of *C. malayense* [48] as well as ethanolic extract of roots of *C. borivilianum* [49]. Saponins from *C. comosum* inhibited tumor promoter effects and exhibited cytotoxicity towards cervical carcinoma HeLa cells, though quantitative data on cytotoxicity were not provided [47]. In seven cell lines originating from solid tumors, *C. malayense* saponin chloromaloside A has been shown to exert a cytotoxic effect that was weaker than that of microtubule destabilizing agent colchicine and topoisomerase II inhibitor ellipticine [48]. One of five spirostane-type saponins purified from *C. borivilianum*, borivilianoside H, was found to be highly toxic to colon tumor cell lines, much less toxic compared to microtubule targeting antitumor compound paclitaxel [49].

Studies published so far have been performed almost exclusively using the underground parts of plants, though secondary metabolite biosynthesis is believed to take place both in leaves and roots based on gene expression profiles [53]. Some of the leaf extracts of *C. borivilianum* obtained with solvents of different polarities exhibited antioxidant properties comparable to those of ascorbic acid and quercetin [54]. Aqueous, methanolic and ethanolic extracts of *C. borivilianum* leaves were shown to be toxic to BHK-21 cells [55] that are derived from normal newborn hamster fibroblasts [56]. Since crude leaf extracts are toxic to normal cells, they can be hardly considered as an efficient anticancer remedy and their biological activities have to be evaluated after further fractionation. Recently, ethanolic root and leaf extracts of *C. comosum* were studied and shown to be toxic to MCF-7, A549 and H1299 cells [57].

Here, we present the study of phytochemistry and biological activities of fractions obtained with serial fractionation of methanolic extract of *C. comosum* leaves. The extract was found to comprise a number of bioactive compounds with known biological activities. *n*-Hexane, chloroform, *n*-butanol and water fractions proved to be substantially different in their content and activity. Chloroform and *n*-butanol fractions possess the majority of antioxidant properties, whereas water fractions exhibited the most promising anti-tumor potential.

## 2. Results

### 2.1. GC-MS Analysis

The GC-MS analysis of methanolic leaf extract of *C.comosum* (Figure 1) revealed nineteen constituents representing 100% of the extract. The detailed tabulations of GC-MS analysis of the extract are given in Table 1. Three major compounds with maximum content included 9,12-octadecadienoic acid (*Z*,*Z*)- (41.27%), neophytadiene (9.57%), and *n*-hexadecanoic acid (7.66%). Sucrose, γ-sitosterol, octadecanoic acid, 9,12-octadecadienoic acid (*Z*,*Z*)-, 2-hydroxy-1-(hydroxymethyl) ethyl ester, stigmasterol, 4′-methylphenyl-1C-sulfonyl-.beta.-d-galactoside, dihydroxyacetone, 3,7,11,15-tetramethyl-2-hexadecen-1-ol, hexadecanoic acid, 2-hydroxy-1-(hydroxymethyl) ethyl ester, and methyl (*Z*)-5,11,14,17-eicosatetraenoate were present at levels of circa 2–4.5% each. The residual compounds detected by GC-MS comprised less than 9% in total.

### 2.2. Phytochemical Screening of Fractions

All fractions of the methanolic extract were shown to contain phenolic compounds, flavonoids, chlorophylls, carotenoids, tannins and reducing sugars (Table 2). The largest amounts of chlorophylls and carotenoids were found in the *n*-hexane and chloroform fractions, whereas phenolic compounds and simple carbohydrates were most abundant in the *n*-butanol fraction and, to a lesser extent, yielded in the chloroform fraction.

In the water fraction, little amounts of phenolic compounds and simple carbohydrates were detected; most of the remaining matter in the fraction was not identified.

### 2.3. Antioxidant Activity of Fractions

The chloroform and *n*-butanol fractions exhibit the highest reducing power and total antioxidant activity (Table 3). The antioxidant properties have to be attributed to phenolic compounds that are most abundant in these fractions.

### 2.4. Cytotoxicity

Human cervical carcinoma cell line HeLa and non-cancerous kidney epithelium cells of *Cercopithecus aethiops* Vero were used for the cytotoxicity studies (Figure 2). Chloroform fraction proved to be the most toxic to cells, with relatively higher (appr. two-fold) toxicity towards HeLa cells. *n*-Hexane and *n*-butanol fractions exhibited similar toxicity to both HeLa and Vero cells. On the other hand, the water fraction was shown to have a very high ratio of Hela to Vero toxicity. While the viability of HeLa cells was efficiently inhibited with IC50 value of 0.12 mg/mL, Vero cells were not affected by the water fraction at doses up to the maximum concentration limited by toxicity of the solvent.

### 2.5. MALDI-TOF Mass Spectrometry of Water Fraction of Methanolic Extract of Leaves of C. comosum

The obtained mass spectra of the water fraction contain signals of different intensities in the range of 500–2200 Da (Figure 3). There were 10 signals with m/z up to 1000 Da, 19 signals in the range of 1200–1700 Da and one signal of 2173 Da detected. The following signals were characterized by the highest intensity: 681 ± 5 Da, 1385 ± 5 Da, 1486 ± 5 Da, 1503 ± 5 Da, and 2173 ± 5 Da. Analysis of mass spectrometry data using the BIOPEP-UWM™ database [60], showed that of water fraction of methanolic extract of leaves of *C. comosum* may contain peptides with antibacterial, antifungal and anticancer activities, as well as enzyme inhibitors that could determine its activity in cell cultures (Table 4).

## 3. Discussion

A number of compounds that were identified in the methanolic extract of leaves of *Chlorophytum comosum* (Green type) have been well characterized.

As is seen from the GC-MS data, unsaturated fatty acids are among the major constituents of the methanolic extract. Linoleic acid (appr. 41% of GC-MS chromatogram area, Figure 1) is considered an essential nutrient with a variety of physiological functions, though there are concerns about the consequences of its excessive consumption for human health. [61,62]. Palmitic acid (7.66%) is one of the most abundant saturated fatty acids in organisms. Its homeostasis is tightly controlled, and its imbalance is related to different physiopathological conditions [63]. Stearic acid (4.32%), synthesized from palmitic acid, is one the most abundant saturated fatty acid in the Western diet. Its effects on cardiometabolic risk markers seem to be different from that of palmitic acid, and a stearic acid-reach diet could be beneficial compared to the ones that are rich with other saturated fatty acids [64,65].

Sucrose (4.5%) is an important nutrient energy source, but its imbalance or excessive consumption has multiple deleterious effects on human health [66,67]. It has been also demonstrated to exert pain preventive effect on infants [68,69].

Some of the constituents of *C. comosum* methanolic leaf extract have been shown to exhibit therapeutic potential. For the second most abundant compound detected with GC-MS, neophytadiene (appr. 10%), anti-inflammatory, antioxidant and cardioprotective activities have been demonstrated [70].

γ-Sitosterol (4.44%) was shown to exert hypocholesterolemic, antidiabetic effects [71].

Stigmasterol (3.09%) exerts antitumor [58,72,73,74], hypolipidemic [75,76,77,78,79,80], antidiabetic [81,82], antimutagenic [83], antiparasitic [84], and antinociceptive [85] effects. It can also alleviate manifestations of osteoarthritis [86,87], inflammation and autoimmune diseases [88,89,90,91], and mitigate central nerve system injuries and disorders in both neuroprotective and function-improving manner [92,93,94,95,96,97,98]. In several studies its antioxidant activity was also demonstrated [79,81,89,92,93,94]. Stigmasterol accumulation, however, can cause cardiac injury [99].

Phytol (2.75%) possesses antioxidant, antimicrobial, anticancer, antidiabetic, hypolipidemic, immunoadjuvant, anti-inflammatory, antimutagenic, antiteratogenic, antinociceptive, antispasmodic, anticonvulsant, anxiolytic, antidepressant activities; it can be used in cosmetics as an active substance or fragrance material [100,101].

5-Hydroxymethylfurfural (1.15%) has been found to exert both deleterious (organotoxic, mutagenic, carcinogenic, pro-oxidant) and beneficial (antihypoxic, anticancer, antioxidant, anti-allergen, antisickling) effects on health [102,103].

In addition to compounds identified by GC-MS, the methanolic leaf extract of *C. comosum* was shown to contain phenolic compounds including tannins that are most abundant in the *n*-butanol fraction (Table 2). Reducing sugars were also found at the highest levels in the *n*-butanol fraction. The chloroform fraction was found to contain two- to three-fold less phenolic compounds and reducing sugars, and *n*-hexane and water fractions had the least amounts of these constituents. Flavonoids were detected at low levels in *n*-hexane and *n*-butanol fractions. The larger part of chlorophylls was extracted from the methanolic extract by *n*-hexane; much lesser chlorophylls were also detected in chloroform fractions. Carotenoids were found in *n*-hexane and chloroform fractions at appr. equal levels.

The antioxidant activity of different fractions of the methanolic leaf extract of *C. comosum* (Table 3) was found to be partially related to the bioactive compound content. The highest reducing power, radical scavenging activity, and total antioxidant activity were found in *n*-butanol and chloroform fractions that correlated with the highest content of phenolic compounds. On the other hand, the reducing power and total antioxidant activity of *n*-hexane and water fractions did not correspond to the phenolic compounds content since water fraction had appr. 1.5-fold higher reducing power and appr. 5-fold lower total antioxidant activity, while both fractions contained similar levels of total phenolics and tannins. It is generally accepted that the antioxidant activity corresponds to the total phenolic content, but in some cases, controversial data were reported suggesting that antioxidant activity might be attributed to antioxidants other than compounds defined as total phenolics [104,105].

The cytotoxicity studies have revealed that the constituents of water fraction have very high toxicity to cancerous HeLa cells compared to non-cancerous Vero cells. In fact, there was no apparent toxicity of water fraction towards Vero cells (Figure 2). The chloroform fraction was shown to be most toxic to cells, but the selectivity towards HeLa cells was significantly lower compared to that of the water fraction.

The data obtained recently with use of *n*-butanol processed ethanolic extracts of defatted roots and leaves of *C. comosum* demonstrated that the extracts were toxic to breast (MCF-7) and lung cancer (A549 and H1299) cell lines but not to the L-132 cell line [57]. Though L-132 (ATCC^®^ CCL-5™) cells cannot be considered non-cancerous since they have signs of HeLa contamination [106,107], these results, together with the data presented herein, suggest that C. *comosum* phytoconstituents can possess promising antitumor activity. Moreover, our data demonstrate that the selective antitumor compounds reside in the fraction obtained after successive removal of *n*-hexane-, chloroform- and *n*-butanol-soluble material.

The total phenolic content of root and leaf ethanolic extracts of *C. comosum* was in good agreement with their antioxidant activity but had poor correlation with cytotoxicity [57]. Our data obtained after more fractionation steps displayed evidence that cytotoxicity of fractions of methanolic leaf extract of *C. comosum* has no correlation with found bioactive compounds or antioxidant activity.

Further considering active substances that can be responsible for the observed cytotoxicity profile of water fraction, one could suggest that, because of negative LogP_o/w_ values, some constituents of the methanolic extract (Table 1) could be present in water fraction. Unfortunately, cytotoxic activity demonstrated so far for that compound does not support any conclusions about an active substance to which the antitumor potential of water fraction could be attributed.

1,3-Dihydroxyacetone, the compound commonly used in sunless tanning products, was shown to be toxic to transformed HEK293T cells [108], A375P melanoma cells [109], and non-cancerous HaCaT cells [110,111]. The experimental settings in the mentioned papers were different, and it is not possible to conclude if dihydroxyacetone could be more toxic towards cancerous than to normal cells. 5-Hydroximethylfurfural, one of the toxicants produced during food processing, exerted moderate cytotoxicity, with IC50 values of millimolar range, to both cancerous and non-cancerous cells [112,113,114].

Glycerol is one of the cryoprotectants that is quite toxic to cells at 37 °C even under short exposure time [115,116], but it exhibits general cytotoxicity regardless of the origin of cells.

Sucrose is well tolerated by cells and affects the protein glycosylation processes rather than the cell viability [117].

The sulfone glycoside 4′-methylphenyl-1C-sulfonyl-β-d-galactoside is quite a rare phytoconstituent. To our knowledge, there is the only MS study that identified 4′-methylphenyl-1C-sulfonyl-β-d-galactoside in Rhodiola Rosea [118]. The biological activity of this compound is not known. The compound was synthesized in a survey of antimalarial agents, but no further studies were published since 1964 [119,120]. A structural analog of the compound with methylphenyl-sulfonyl moiety bound to the oxygen of the 6C atom, inhibited the sugar uptake process in bacteria [121].

Methyl 6-o-[1-methylpropyl]-β-d-galactopyranoside was found as one of four most abundant constituents in the ethanolic leaf extract of the Kofat cultivar of Catha edulis. The cytotoxicity studies did not reveal any selectivity in toxicity of the extracts towards cancer cells compared to normal fetal lung fibroblast [122].

MALDI-TOF mass spectrometry of the water fraction has revealed a number of compounds that may belong to bioactive peptides. The EQRPR pentapeptide, identified in rice bran, was shown to be toxic to several cancer cell lines, though at relatively high concentrations [123]. The peptide FFVAPFPEVFGK and its close analogs were found in skin extract of amphibians and hydrolysates of casein [124]. It was found to inhibit several proteinases involved in cancer progression though it did not exhibit cytotoxicity towards cancer cells [125]. The KWKLFKKIKKIKFLHSAKKF peptide, corresponding to CancerPPD database data [126], possesses mild cytotoxic activity against several cancer cell lines with IC50 values ranging at 65–100 µM. At the same time, this peptide is an artificial chimera of two antibacterial peptides of Hyalophora cecropia and Xenopus laevis [127], hence it is hardly possible that this compound is naturally produced by *C. comosum*.

Thus, the water fraction of the methanolic extracts of leaves of *C. comosum* contains compound(s) that have highly selective toxicity towards cancer cells. This anti-cancer cytotoxicity is apparently unrelated to antioxidant activity. Active compounds of the water fraction may be presented by both small molecules and bioactive peptides.

## 4. Materials and Methods

### 4.1. Chemicals

Methyl alcohol 99.9%, *n*-hexane 95%, chloroform 99%, *n*-butanol 99%, dimethyl sulfoxide (DMSO) 99.7%, aluminum chloride 99.9%, sodium acetate >99%, quercetin, acetone >99.5%, picric acid ≥98.0%, dextrose 99.5%, sodium carbonate 99.5%, phosphate buffered saline (pH 7.2 at 25 °C), potassium ferricyanide (III), trichloroacetic acid, α-Cyano-4-hydroxycinnamic acid (for MALDI-TOF MS), acetonitrile ≥99.0%, iron (III) chloride 99%, ascorbic acid ≥99%, sulfuric acid 99.9%, sodium phosphate 96%, ammonium molybdate, potassium persulfate ≥99.0%, 2,2′-azino-bis(3-ethylbenzothiazoline-6-sulfonic acid) diammonium salt (ABTS), 6-hydroxy-2,5,7,8-tetramethylchroman-2-carboxylic acid (Trolox) 97%, Folin–Denis’ reagent, gallic acid, tannin and chemical standards were purchased from Sigma-Aldrich.

EMEM and DMEM incubation media, L-glutamine, essential amino acids mixture, penicillin-streptomycin 100x lyophilized mixture, 0.25% trypsin solution and 0.02% versene solution were purchased from PanEco, Ltd. (Moscow, Russia), fetal bovine serum—from BioWest (Nuaillé, France), and MTT (3-(4,5-dimethylthiazole-2-yl)-2,5-diphenyl-tetrazolium bromide)—from Diaem, Ltd. (Moscow, Russia).

### 4.2. Plant Material

*Chlorophytum comosum* (Green type) (Thunb.) Jacques was kindly provided by the Botanical Garden of the North-Caucasus Federal University. Vegetative propagation of the plant was carried out in the Greenhouse of the Moscow Region State University. The plant was harvested in August 2019.

### 4.3. Sample Preparation

Freshly cut leaves were washed under running tap water for 30 s, chopped with scissors to pieces of 1–2 cm and subjected to freeze-drying. The material was frozen in SE10-45 (TEFCOLD, Viborg, Denmark) at a temperature of minus 39.0–40.0 °C for 72 h. Subsequent drying was carried out for 27–30 h in a light-protected chamber in LS-500 freeze dryer (Prointech, Puschino, Russia) with average working pressure in the dryer chamber of 90–100 Pa, the condenser temperature minus 47–49 °C. The residual moisture content of the dried material amounted to 5–6% as was measured using an Ohaus MB 25 moisture meter (Ohaus Corporation, Parsippany, NJ, USA) at a temperature of 100 °C in an automatic mode.

The lyophilized leaves were ground to a powder with particle size of 1 mm or less using a VT-1541 VK grinder (Vitek, SuZhou, China). The powdered samples were stored at 4 °C in a dark hermetically sealed container until testing for no more than one week.

### 4.4. Extraction and Fractionation

Of the powdered lyophilized leaves, 42.0 g was extracted with 420.0 mL of methanol for 14 days at room temperature (23 ± 0.5 °C). The extract was filtered through cheesecloth and paper filters and then dried using a rotary evaporator RV 10 Basic V (IKA, Germany) at a temperature of 45–50 °C until complete solvent removal [128,129]. The yield of dry matter was 7.0 ± 0.1 g.

The methanolic extract was subjected to fractionation according to Singh et al. (2009) [130] with slight modifications. The extract (7.0 g) was dissolved in a methanol:water (9:1) mixture and the solution was successively divided into fractions with *n*-hexane, chloroform and *n*-butanol using a separatory funnel. Organic solvents were removed using a rotary evaporator, water from the final aqueous fraction was removed by lyophilization. The resulting dried pellets were dissolved in DMSO to a concentration of 50 mg/mL for *n*-hexane fraction and 100 mg/mL for chloroform, *n*-butanol and water fractions.

### 4.5. Gas Chromatography-Mass Spectrometry (GC-MS) Analysis of Methanolic Leaf Extract

Of the powdered lyophilized leaves, 5 g was extracted with 100 mL of methanol for 24 h at room temperature (23 ± 0.5 °C). The extract was filtered through 0.22 μm PTFE syringe filters (Merck Millipore, Burlington, MA, USA).

For GC-MS analysis, a GCMS-QP2010 Ultra (Shimadzu, Japan) system with a DB-5MS capillary column (5% diphenyl – 95% dimethylsiloxane copolymer, 60 m long, inner diameter 0.25 mm, film thickness of the stationary phase 0.25 μm) was used. The carrier gas (helium) flow rate was 1.38 mL/min. The temperature program was as follows: isothermal for 4 min at 100 °C, increased at 15 °C/min to 300 °C, isothermal for 26 min. The ionization voltage was 70 eV. The scanning speed was 3333 s over the 30–550 *m*/*z* range. For qualitative analysis of the essential oil components, the NISTO.5a mass spectra library was used.

### 4.6. Matrix-Assisted Laser Desorption/Ionization (MALDI) Time-of-Flight (TOF) Mass Spectrometry

Water fraction was subjected to proteomic analysis using MALDI-TOF mass spectrometry. The solution was centrifuged at 7000 g for 4 min. The supernatant (1 microliter) was deposited on the MALDI plate. Pretreated and untreated samples were overlaid with 1 microliter of matrix solution (saturated solution of α-cyano-4-hydroxycinnamic acid in 50% acetonitrile and 2.5% trifluoroacetic acid). The matrix sample was cocrystallized by air drying at room temperature. Measurements were performed with a Microflex mass spectrometer (Bruker Daltonik, Bremen, Germany) using Daltonics FlexControl software (version 3.3.64). Spectra were recorded in the positive linear mode (laser frequency, 60 Hz; ion source 1 voltage, 19.4 kV; ion source 2 voltage, 17.3 kV; lens voltage, 9.1 kV; mass range, 0 to 20,000 Da). The internal calibration was performed using the mass test standard MBT (Bruker Daltonics, Bremen, Germany). For each spectrum, 4000 shots from different positions of the target spot (automatic mode) were collected and analyzed. Protein identification was performed using the BIOPEP-UWM™ database [60].

### 4.7. Determination of Total Phenolics

The total phenolic content was measured according to Swain and Hillis method [131]. Then, 100 μL of a DMSO-dissolved fraction was mixed with 0.5 mL of the Folin–Denis’ reagent. After 3 min, 1 mL of 20% Na_2_CO_3_ was added, and volume of the mixture was brought to 10.0 mL with distilled water. After incubation in the dark for 90 min, absorbance of the resulting blue complex was measured at 750 nm using an SF-102 spectrophotometer (NPO Interfotofizika, Moscow, Russia). Gallic acid was used as a standard. The total phenolic content was expressed in mg equivalent of gallic acid per 1 mL of sample (mg GAEs/mL).

### 4.8. Determination of Total Tannins

The total tannin content was determined according to the Price and Butler method [132] with modifications. A DMSO dissolved extract was diluted 500-fold with distilled water to a concentration of 100–200 μg/mL. An aliquot of 0.5 mL of diluted sample was added to 1 mL of 1% FeCl_3_, then the volume was adjusted to 10 mL with distilled water. After 5 min incubation at room temperature, the optical density was measured at 720 nm. Tannin was used as a standard.

### 4.9. Determination of Total Flavonoids

The flavonoid content was studied with the aluminum chloride colorimetric method [133] with slight modifications. An aliquot of 50 μL of DMSO dissolved extract was mixed with 1.5 mL of ethanol, 100 μL of 10% aluminum chloride, 100 μL of 1M sodium acetate and 2.8 mL of distilled water. The resulting mixture was incubated at room temperature for 30 min in the dark, then absorbance at 415 nm was read. Quercetin was used as a standard. The total flavonoid content was expressed in mg equivalent of quercetin per 1 mL of sample (mg QEs/mL).

### 4.10. Determination of Chlorophyll a, Chlorophyll b, and Total Carotenoid

The content of chlorophylls a and b and total carotenoid was measured spectrophotometrically. The DMSO dissolved samples were diluted 1000-fold with 80% acetone and clarified with centrifugation at 10,000× *g* at 4 °C for 15 min. The absorbance of the supernatant was measured at 470 nm, 646.8 nm, and 663.2 nm, and the concentrations of chlorophyll a, chlorophyll b and carotenoids (μg/mL) were calculated using equations for 80% acetone as described [134].

### 4.11. Determination of Carbohydrates

Reducing sugars were measured with the use of alkaline picric acid method [135,136]. In brief, 1 mL of 1% picric acid was added to 0.5 mL of a test fraction, then 3 mL of 20% sodium carbonate was added. Samples were boiled for 30 min, cooled and adjusted to 10.0 mL by distilled water. The absorbance was read at a wavelength of 460 nm. Glucose was used as a standard.

### 4.12. Ferric Reducing Power Assay

The assay was carried out as described [137,138] with slight modifications. As such, 10 μL of DMSO dissolved fractions was mixed with 2 mL of 0.2 M sodium phosphate buffer, pH 6.6, and 1 mL of 1% potassium ferricyanide. The resulting mixture was incubated at 50 °C for 20 min. The reaction was terminated by addition of 1 mL of 10% trichloroacetic acid. The mixture was centrifuged at 3000× *g* for 10 min, then 0.5 mL of freshly prepared 0.1% FeCl_3_ was added to the supernatant, and the absorption was measured at 700 nm. Ascorbic acid was used as a standard. The reducing power was expressed in mg equivalent of ascorbic acid per 1 mL of sample (mg AAEs/mL).

### 4.13. Assessment of Total Antioxidant Activity (TAA)

The total antioxidant activity was assayed with the use of the phosphomolybdenum method [138,139]. As such, 50 μL of DMSO dissolved fractions was mixed with 4 mL of the reagent solution (0.6 M sulfuric acid, 28 mM sodium phosphate and 4 mM ammonium molybdenum). The mixture was incubated at 95 °C for 90 min, then cooled to room temperature. The absorbance was measured at 695 nm. Ascorbic acid was used as a standard. The TAA was expressed in mg equivalent of ascorbic acid per 1 mL of sample (mg AAEs/mL).

### 4.14. Trolox Equivalent Antioxidant Capacity (TEAC) Assay

The assay was carried out according to described by Piskov et al. (2020), Rzhepakovsky et al. (2021) method [140,141]. 2,29-azinobis-(3-ethylbenzothiazoline-6-sulfonic acid) (ABTS) was dissolved in water to a concentration of 7 mM. The ABTS^•+^ cationic radical generation was initiated by addition of 1 mL of 14.7 mM potassium persulfate to 5 mL of ABTS. The resulting mixture was kept in the dark at room temperature for 24 h before use. To perform the assay, the ABTS solution was diluted with distilled water to an optical density of 0.70 (±0.02) at 734 nm, then 5 μL of DMSO dissolved fractions were added to 2 mL of ABTS solution. The absorbance at 734 nm was measured 3 min after mixing. Trolox solution at a concentration of 1 mM was used as a standard. The antioxidant capacity was expressed in mg equivalent of Trolox per 1 mL of sample (mg TEs/mL).

### 4.15. Cytotoxicity Assay

Cancer cells HeLa (human cervical epithelioid carcinoma cell line, subclone M) and non-cancerous Vero cells (African green monkey renal epithelial cells) were purchased from the Russian Cell Culture Collection (Institute of Cytology RAS, St., Russia). The HeLa and Vero cells were grown in EMEM and DMEM media, respectively. The incubation media were supplemented with 10% fetal bovine serum, penicillin (100 U/mL) and streptomycin (100 µg/mL). The cells were cultured at 37 °C in a humidified atmosphere of 5% CO_2_. Cytotoxicity of *C. comosum* fractions was studied with the use of the MTT staining method [142]. Cells were plated in 96-well plates (5 × 10^3^ cells/well). Then, 24 h after plating, growth medium was replaced by medium containing different quantities of fractions dissolved in DMSO. The final concentration of DMSO did not exceed 1%. Then, 24 h after addition of samples, 10 µL of MTT stock solution (5 mg/mL) was added into each well and plates were incubated for 4 h. The media were then removed and MTT formazan was dissolved in DMSO (100 µL/well). The absorption of the samples was read at 570 nm (reference wavelength 650 nm). MTT staining of control samples treated with DMSO was taken as 100%.

## 5. Conclusions

Our data demonstrate that methanolic extract of leaves of *Chlorophytum comosum* (Green type) (Thunb.) Jacques can be a source of a number of bioactive compounds including those that can be used as dietary supplements as well as constituents with broad biological activity and therapeutic potential. We have found that water fraction of the methanolic extract obtained after successive removal of *n*-hexane, chloroform and *n*-butanol soluble matter comprises compound(s) with high selectivity of cytotoxicity towards cancer cells. The exact nature of this antitumor potential has to be elucidated.

## Figures and Tables

**Figure 1 molecules-27-00762-f001:**
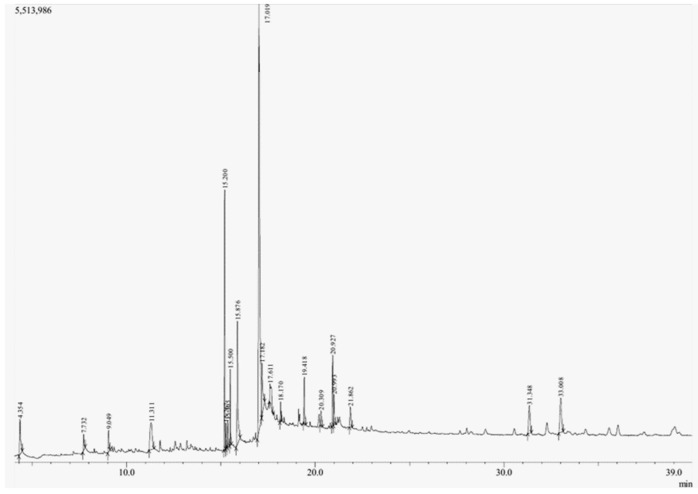
GC-MS chromatogram of the methanolic extract of leaves of *Chlorophytum comosum* (Green type).

**Figure 2 molecules-27-00762-f002:**
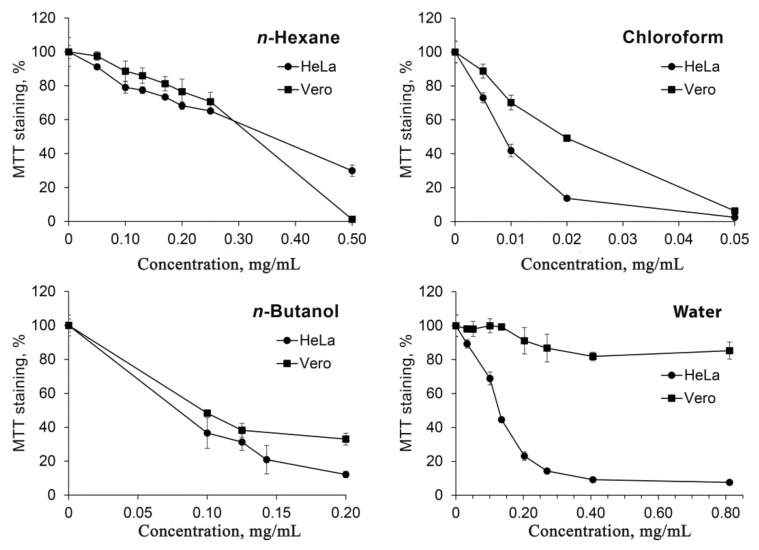
Cytotoxicity of fractions of methanolic extract of leaves of *Chlorophytum comosum (Green type)*.

**Figure 3 molecules-27-00762-f003:**
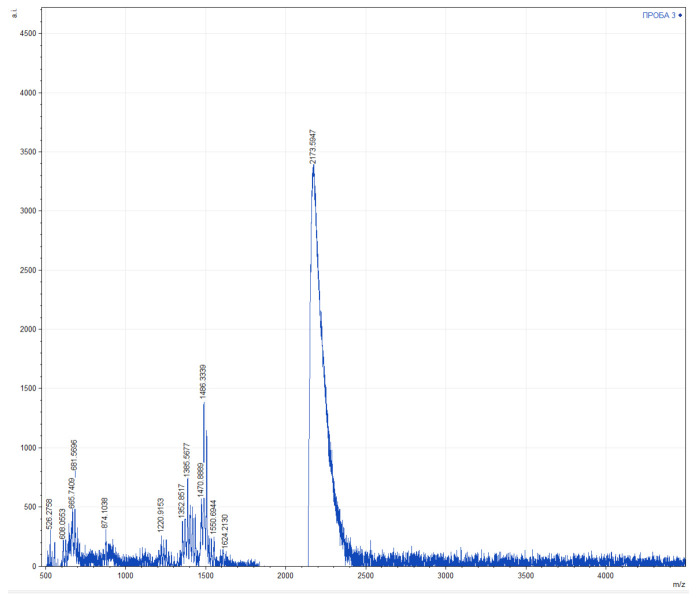
Data of the MALDI-TOF mass spectrometry of water fraction of methanolic extract of leaves of *Chlorophytum comosum* (Green type).

**Table 1 molecules-27-00762-t001:** Phytoconstituents detected in the methanolic extract from leaves of *Chlorophytum comosum* (Green type).

Peak No.	RT	Name of the Phytoconstituents	Structure	Mol. Formula	Molecular Weight	Peak Area %	LogP_o/w_	Chemical Class
1	4.354	1,3-Dihydroxyacetone	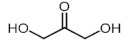	C_3_H_6_O_3_	90	2.85	−1.95	Saccharide
2	7.732	Glycerol	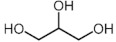	C_3_H_8_O_3_	92	1.01	−1.93 *	Polyol
3	9.049	5-Hydroxymethylfurfural	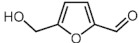	C_6_H_6_O_3_	126	1.15	−0.17 *	Aromatic aldehyde
4	11.311	Sucrose	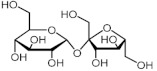	C_12_H_22_O_11_	342	4.5	−2.63 *	Disaccharide
5	15.2	7,11,15-trimethyl-3-methylidenehexadec-1-ene (Neophytadiene)	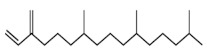	C_20_H_38_	278	9.57	8.12 *	Isoprenoid hydrocarbon
6	15.267	2-Hexadecene, 3,7,11,15-tetramethyl-, [R-[R*,R*-(E)]]- (2-Phytene)	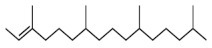	C_20_H_40_	280	1.15	8.81 *	Isoprenoid hydrocarbon
7	15.365	E-6-Octadecen-1-ol acetate	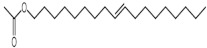	C_20_H_38_O_2_	310	1.57	8.34 *	Fatty alcohol ester
8	15.5	3,7,11,15-Tetramethyl-2-hexadecen-1-ol (Phytol)	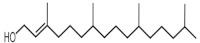	C_20_H_40_O	296	2.75	7.89 *	Diterpene alcohol
9	15.876	n-Hexadecanoic acid(Palmitic acid)	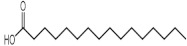	C_16_H_32_O_2_	256	7.66	7.17	Fatty acid
10	17.019	9,12-Octadecadienoic acid (*Z*,*Z*)- (Linoleic Acid)	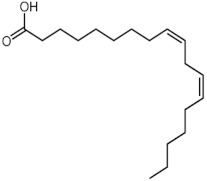	C_18_H_32_O_2_	280	41.27	7.05 [58]	Fatty acid
11	17.182	Octadecanoic acid (Stearic acid)	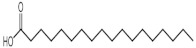	C_18_H_36_O_2_	284	4.32	8.23 [58]	Fatty acid
12	17.611	4′-Methylphenyl-1C-sulfonyl-β-d-galactoside	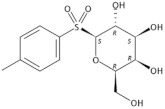	C_13_H_18_O_7_S	318	2.88	−1.21 *	Sulfoneglycoside
13	18.17	cis-7-Dodecen-1-yl acetate (Looplure)	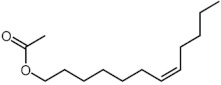	C_14_H_26_O_2_	226	0.8	5.66 *	Fatty alcohol ester
14	19.418	Hexadecanoic acid, 2-hydroxy-1-(hydroxymethyl) ethyl ester (2-Palmitoylglycerol)	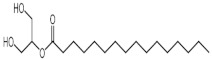	C_19_H_38_O_4_	330	2.24	5.77 *	Fatty acid ester
15	20.309	Methyl 6-o-[1-methylpropyl]-β-d-galactopyranoside	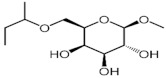	C_11_H_22_O_6_	250	2.66	−0.30 *	Monosaccharide
16	20.927	9,12-Octadecadienoic acid (*Z*,*Z*)-, 2-hydroxy-1-(hydroxymethyl) ethyl ester (2-Linoleoyl Glycerol)	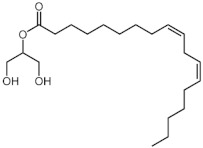	C_21_H_38_O_4_	354	4.04	5.55 *	Fatty acid ester
17	20.993	Methyl (Z)-5,11,14,17-eicosatetraenoate	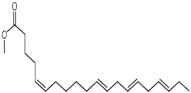	C_21_H_34_O_4_	318	2.05	6.82 *	Fatty acid ester
18	31.348	Stigmasterol	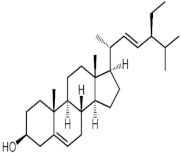	C_29_H_48_O	412	3.09	6.95 *	Stigmastane
19	33.008	γ-Sitosterol	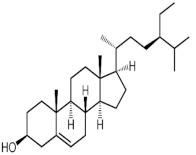	C_29_H_50_O	414	4.44	7.27 *	Stigmastane

*—Calculated with ALOGPS software [59].

**Table 2 molecules-27-00762-t002:** Total bioactive compounds found in fractions of the methanolic extract of leaves of *Chlorophytum comosum* (Green type), mean ± SD (*n* = 10).

Bioactive Compounds	*n*-Hexane Fraction	Chloroform Fraction	*n*-Butanol Fraction	Water Fraction
Relative content of dry matter, %	19.9 ± 0.5	22.8 ± 0.57	23.4 ± 0.4	33.9 ± 0.7
Concentration of extracted matter in DMSO, mg/mL	50 ± 2.0 *	100 ± 2.0	100 ± 3.0	100 ± 3.0
Total phenolic content (TPC), mg GAE/mL	1.08 ± 0.02	5.6 ± 0.1	19.9 ± 0.5	0.51 ± 0.02
Tannins, mg/mL	1.73 ± 0.04	4.22 ± 0.1	15.75 ± 0.4	1.77 ± 0.03
Total flavonoid content (TFC), mg QE/mL	0.05 ± 0.001	ND **	0.09 ± 0.001	ND
Chlorophyll a, mg/mL	3.16 ± 0.05	0.16 ± 0.004	ND	ND
Chlorophyll b, mg/mL	1.77 ± 0.02	0.26 ± 0.01	ND	ND
Carotenoids, mg/mL	0.74 ± 0.02	0.62 ± 0.02	ND	ND
Reducing sugars, mg/mL	1.53 ± 0.03	9.18 ± 0.1	22.2 ± 0.5	1.28 ± 0.02

* The extracted matter in the *n*-hexane fraction had low solubility in DMSO. ** ND, not detected.

**Table 3 molecules-27-00762-t003:** Antioxidant activity of the fractions of methanolic extract from leaves of *Chlorophytum comosum* (Green type)*,* mean ± SD (*n* = 10).

Antioxidant Activity Criterion	*n*-Hexane Fraction	Chloroform Fraction	*n*-Butanol Fraction	Water Fraction
Reducing power, mg AAE eq/mL	0.39 ± 0.01	6.25 ± 0.2	33.1 ± 2.5	0.58 ± 0.07
ABTS radical scavenging activity, mg TEs/mL	0.27 ± 0.01	0.36 ± 0.01	0.81 ± 0.05	0.27 ± 0.02
Total antioxidant activity (TAA), mg AAE eq/mL	9.7 ± 0.32	17.4 ± 0.6	35.9 ± 4.5	1.8 ± 0.9

**Table 4 molecules-27-00762-t004:** Characterization of the proteomic analysis of water fraction of methanolic extract of leaves of *Chlorophytum comosum* (in accordance with the BIOPEP-UWM™ database).

Chemical Mass, Da	ID	Sequence	Activity	Int.
681	-	-	-	730
-	8252	EQRPR	anticancer	-
1385	-	-	-	675
-	3194	FLPAIAGILSQLF~	hemolytic	-
-	8311	FFVAPFPEVFGK	anticancer	-
-	9291	KKLFKKILKKL~	antifungal	-
-	9466	FKCRRWQWR	antibacterial	-
1486	-	-	-	1335
	2979	KKAVRRQEAVDAL	CaMKII inhibitor	-
-	2989	KKALRRDEAVDAL	CaMKII inhibitor	-
-	2990	KKALRRNEAVDAL	CaMKII inhibitor	-
-	2991	KKALRRQEGVDAL	CaMKII inhibitor	-
-	3190	SSSKEENRIIPGGI	antibacterial	-
-	5450	GLFDAIGNLLGGLGLG	antibacterial	-
-	5474	GLFDIVKKIAGHIA	antibacterial	-
-	8335	SDIPNPIGSENSEK	antibacterial	-
-	9458	RWQWRWQWR	antibacterial	-
-	9783	GEHGGAGMGGGQFQPV	alpha-amylase inhibitor	
-	9784	GEHGGAGMGGGQFQPV	pancreatic lipase inhibitor	
	9785	GEHGGAGMGGGQFQPV	lipoxygenase inhibitor	
	9786	GEHGGAGMGGGQFQPV	cyclooxygenase-1 inhibitor	
	9787	GEHGGAGMGGGQFQPV	cyclooxygenase-2 inhibitor	
1503	-	-	-	1083
-	2987	KKALRREEAVDAL	CaMKII inhibitor	-
-	3006	KKALYRQEAVDAL	CaMKII inhibitor	-
-	3008	KKALRYQEAVDAL	CaMKII inhibitor	-
-	3920	GLFDIIKKIAESIG	antibacterial	-
-	5454	GLFDIIKKIAESIG	antibacterial	-
-	5457	LDIVKKVVGAFGSLG	antibacterial	-
-	9294	WKLFKKILKWL~	antifungal	-
-	9295	WKLFKKILKWL~	hemolytic	-
-	9313	WKLFKKILKKLG	antifungal	-
-	9314	WKLFKKILKKLG	hemolytic	-
2173	-	-	-	2403
	3822	GLLRRLRKKIGEIFKKYG	antibacterial	-
	7053	KWKLFKKIKFLHSAKKF	anticancer	-

## Data Availability

Data of the research is available is on request to the corresponding author.

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
