# Peer review of "Phytochemical Characterization, Antioxidant Activity, and Cytotoxicity of Methanolic Leaf Extract of *Chlorophytum Comosum* (Green Type) (Thunb.) Jacq"

_molecules, 2022, doi:10.3390/molecules27030762_

Round 1

Reviewer 1 Report

This is one of the routine research findings on 'Safed Museli' as lot of published research work is available on phytochemical constituents of roots of 'Safed Museli' and their immunomodulatory, adaptogenic, aphrodisiac and other beneficial effects on human health is already available. The authors tried to explore the possibility of utilizing foliage of Chlorophytum (though some reports are already available) for their health promoting benefits but comparative results on relative concentrations of important phytochemicals and secondary metabolites of leaves and roots might have been given better picture and purpose of the study. Discussion can be condensed.

-The authors have carried out elaborate phytochemical characterization, antioxidant activity and cytotoxicity of methanolic leaf extract of Chlorophytum comosum (Green type) (Thunb.) Jacq to find out important phytochemicals and their activities for their ultimate use as medicine and/or dietary supplements.

-Abstract, line 21: Twenty nineteen compounds belonging to

-There are some issues related to the use of dried pellets for preparation of DMSO dissolved fraction for the estimation of Phenols, flavonoids, chlorophylls, tannins and carotenoids. I am not quite sure about the sample preparation and estimation of these constituents using DMSO extract (extraction, fractionation, drying) as mentioned in the text.

-(Barabanov et al., 2018)- missing in the reference portion

-4.7. Determination of Total Phenolics, second last line of the para: total fenolics content must be replaced with total phenolics content

- 2. Results 2.1. GC-MS Analysis The GC-MS analysis of methanolic leaf extract of C. comosum (Figure 1) revealed twenty nineteen constituents representing 100% of the extract

-Discussion needs to be modified to make it more interesting and crisper.

-Uniform referencing style and their mention in text has not been followed

-This is one of the routine research findings on 'Safed Museli' as lot of published research work is available on phytochemical constituents of roots of 'Safed Museli' and their immunomodulatory, adaptogenic, aphrodisiac and other beneficial effects on human health is already available. The authors tried to explore the possibility of utilizing foliage of Chlorophytum (though some reports are already available) for their health promoting benefits but comparative results on relative concentrations of important phytochemicals and secondary metabolites of leaves and roots might have been given better picture and purpose of the study. The utility of this research work could have been better expressed if there was comparative phytochemical profiling of roots as well as leaves.

Author Response

Dear Reviewer,

Thank you very much fro you attention to our work and kind recommendations which made our article better. 

Please see Respond to reviewer document in attachment

Author Response

(The authors gave the same response as above.)

Reviewer 3 Report

The manuscript entitled "Phytochemical Characterization, Antioxidant Activity and  Cytotoxicity of Methanolic Leaf Extract of Chlorophytum comosum (Green type) (Thunb.) Jacq" evaluated the methanolic extract of leaves of Chlorophytum comosum (Green type) (Thunb.) Jacques. The research is valuable, but the research on the relevant mechanism is not in-depth.

Author Response

(The authors gave the same response as above.)

Round 2

Reviewer 3 Report

Accept.